# Upregulated Immunogenic Cell-Death-Associated Gene Signature Predicts Reduced Responsiveness to Immune-Checkpoint-Blockade Therapy and Poor Prognosis in High-Grade Gliomas

**DOI:** 10.3390/cells11223655

**Published:** 2022-11-17

**Authors:** Xin Tang, Dongfang Guo, Xi Yang, Rui Chen, Qingming Jiang, Zhen Zeng, Yu Li, Zhenyu Li

**Affiliations:** 1Department of General Internal Medicine, Chongqing University Cancer Hospital, Chongqing 400030, China; 2Department of Pathology, Chongqing University Cancer Hospital, Chongqing 400030, China; 3Department of Clinical Laboratory, Chongqing University Cancer Hospital, Chongqing 400030, China; 4Chongqing Cancer Multi-Omics Big Data Application Engineering Research Center, Chongqing University Cancer Hospital, Chongqing 400030, China; 5Chongqing Key Laboratory of Translational Research for Cancer Metastasis and Individualized Treatment, Chongqing University Cancer Hospital, Chongqing 400030, China

**Keywords:** glioma, immunogenic cell death, immune microenvironment, prognosis, immunotherapy

## Abstract

**Background:** Immunogenic cell death (ICD) has emerged as a potential mechanism mediating adaptive immune response and tumor immunity in anti-cancer treatment. However, the signature of ICD in high-grade gliomas (HGGs) remains largely unknown, and its relevance to immunotherapies is still undetermined. The purpose of this study is to identify ICD-associated genotypes in order to explore their relevance to tumor immunity, patient prognosis and therapeutic efficacy of immune checkpoint blockade (ICB) therapy in HGGs. **Methods:** Bulk RNA-seq data and clinical information on 169 and 297 patients were obtained from the Cancer Genome Atlas (TCGA) and China Glioma Genome Atlas (CGGA), respectively. The functional enrichment and characterization of ICD genotyping were detected, and the ICD prognostic signature prediction model was constructed using least absolute shrinkage and selection operator (LASSO) regression. The responsiveness to immunotherapy was predicted according to the scoring of the ICD prognostic signature. **Results:** The HGG patients with high ICD gene signature (C1) showed poor outcomes, increased activity of immune modulation and immune escape, high levels of immune-checkpoint markers, and HLA-related genes, which may explain their reduced response to ICB immunotherapy. A gene set of the ICD signature, composing *FOXP3*, *IL6 LY96*, *MYD88* and *PDIA3*, showed an independent prognostic value in both the TCGA and the CGGA HGG cohort. **Conclusions:** Our in silico analyses identified the ICD gene signature in HGGs with potential implications for predicting the responsiveness to ICB immune therapy and patient outcomes.

## 1. Introduction

High-grade diffuse gliomas in adults (HGGs), composing astrocytoma, IDH-mutant grade 3–4, oligodendroma, IDH-mutant and 1p/19q co-deletion, grade 3, and glioblastoma (GBM), IDH-wild type, grade 4, are the most aggressive malignant tumors of the central nervous system, with dismal outcomes [1,2]. The standard-of-care treatment for HGGs, including chemotherapy, radiotherapy, and maximized surgical resection, show limited efficacy in improving HGG patients’ outcomes. Immune therapy, including immune-checkpoint-blockade (ICB) therapy, has shown promising therapeutic effects in several malignant tumors but not in HGGs [3]. Accumulating data indicates that the high infiltration of suppressive immune cells, including regulatory T-cell and tumor-associated macrophages, poses great challenges for ICB treatment in HGGs [4,5]. The investigation of the mechanism underlying tumor-immunity dysfunction in HGGs is a prerequisite for the development of effective immune therapy.

Immunogenic cell death (ICD) represents a functionally unique response pattern after induction by organellar or cellular stress, specifically associated with apoptotic cell death, and leads to the passive release of numerous damage-associated molecular patterns (DAMPs) in a spatio-temporal manner [6,7,8].

There is increasing evidence showing that ICD is a prognostic factor associated with long-term survival in cancer patients [9,10,11,12]. Reinforcing anti-cancer immune response by ICD induction is emerging as a potential therapeutic option in cases of resistance to conventional chemo- or radio-based treatment [13,14,15]. Thus, the evaluation of ICD activity in tumors is important for therapeutic-efficacy prediction. Tumor cells with pre-existing therapy-resistant variants pose a crucial challenge to the therapeutic use of ICD inducers and ICD-associated danger signaling [16]. Further treatments are needed based on combinations of ICD inducers that can be applied simultaneously in order to reduce the probability of resistance arising. A recent work by Garg AD. et al. [17] identified a 33-metagene signature of ICD associated with improved survival of patients with lung, breast or ovarian tumors [18]. However, HGGs showed significant tumor heterogenicity, and an ICD evaluation method for HGGs has not been reported.

In this study, we aimed to demonstrate ICD-associated biomarkers and constructed an ICD-signature gene set to identify its association with tumor immunity in HGGs, and response to immune checkpoint blockade (ICB) immunotherapy in HGGs. We identified an ICD-related genotyping method that may reflect ICD signatures and had a potential role in predicting poor prognosis in patients with HGGs. Our in silico analyses identified the ICD gene signature in the HGGs with potential implications for predicting the responsiveness to ICB immune therapy and patient outcomes.

## 2. Material and Methods

### 2.1. Datasets

The gene-expression spectrum and single-nucleotide-mutation data of 169 HGG samples were obtained from The Cancer Genome Atlas (TCGA, https://cancergenome.nih.gov/, accessed on 1 September 2022). RNA gene expression profiles of 325 HGG samples were obtained from the China Glioma Genome Atlas project (CGGA, http://www.cgga.org.cn/, accessed on 1 September 2022) [19]. The follow-up information of 159 patients from TCGA and 297 patients from CGGA was obtained after excluding incomplete clinical information. The details were showed in Table 1.

### 2.2. Consensus Clustering of ICD in HGGs

The “ConcensusClusterPlus” tool in R software was utilized to perform consensus clustering using ICD gene expression and follow-up information of the patients after univariate Cox regression analysis [17,20]. Subsequently, the ideal cluster numbers between k = 2 to 9 were evaluated repeatedly, and the results were stable. The most appropriate ICD genotyping was found according to the strength of gene-expression correlation. The cluster map was created by pheatmap tool of R software (version 4.0.3, Hadley Wickham, Auckland, New Zealand).

### 2.3. Identification of Differentially Expressed Genes (DEGs), Gene Ontology (GO) and Kyoto Encyclopedia of Genes and Genomes (KEGG) [21,22]

DEGs were analyzed between C1 and C2 groups. The cut-off criteria for DEGs were determined as adjusted *p*-value < 0.01 and | logFC | > 2 of false discovery rate (FDR). DEGs of C1 and C2 groups were employed to evaluate GO and KEGG pathways using the “clusterProfiler” package, “org.hs.eg. db” package, “Enrichplot” package, “GOplot” package, and “GGplot2” package of R software.

### 2.4. Characterization of Immune Landscape and Correlation [23]

Immune and stromal cell scores were calculated after applying the Estimate algorithm in “estimate” and “limma” of R-package to predict tumor purity and the presence of infiltrating stromal/immune cells in glioma tissues. The immune cell content of each glioma sample was calculated using “e1071” and “preprocessCore” of R packages. The relationship between ICD genotyping and immune function was evaluated by comparing the content of immune cells, the expression levels of HLA family genes and immune checkpoint-related genes between ICD genotypings.

### 2.5. Construction of ICD Prognostic Signature Model and Survival Analysis [24,25]

Univariate cox regression analysis was used to identify ICD genes with prognostic value. Least absolute shrinkage and selection operator (LASSO) Cox regression analysis was performed for prognostic ICD genes using the “glmnet” package. Multivariate Cox regression analysis was used to identify ICD genes with independent prognostic value. The optimal-related gene prognostic markers were selected according to the lowest Akaike Information Standard (AIC) value. The Risk Score of each patient was calculated according to the following formula: Risk Score = ∑n k = 0 coef (k) * X (k), COEF (k) and X (k) represent regression coefficients, and each value represents ICD gene. Kaplan–Meier survival curves were drawn to compare the OS or PFS of glioma patients with different ICD gene prognostic risk scores. The receiver operating characteristic (ROC) curve was calculated by “survivalROC” of R package to show the prognostic value of ICD gene prognostic risk score at 1, 3 and 5 years, as well as other clinical characteristics (age, sex, pathological grade, IDH mutation, 1p19q co-deletion, MGMT methylation).

### 2.6. Prediction of Response to ICB Immunotherapy [26,27]

“pRRophetic” of R package was used to screen potential sensitive drugs of glioma based on ICD prognostic signature model. Spearman correlation analysis was applied to detect the correlation between IC50 and ICD prognostic risk score. The significance of IC50 between two ICD prognostic risk score groups was detected. The expression of glioma samples spectrums was imported into Tumor Immune Dysfunction and Exclusion platform (TIDE, http://tide.dfci.harvard.edu/, accessed on 10 September 2022), and then, the TIDE rating of each sample was calculated; subsequently, the TIDE score difference between two ICD prognostic risk score groups was compared. On the TIDE web, predicted responder predictions by the threshold of the TIDE score were set by a user (default is 0). The TIDE score means “response” to ICB when it represents positive value; otherwise, “no response”. ICB therapy, including anti-PD1 and anti-CTLA4 strategies, was employed in this study as representative immunotherapy. T cell dysfunction and low level of cytotoxic T lymphocyte (CTL) was used to predict the ICB immunotherapy response [18,28]. In the CTL-high tumors, TIDE correlates with T cell dysfunction signature and predicts non-responders with high scores of T cell dysfunction [29].

### 2.7. Statistical Analysis

All data were statistically analyzed using R software for non-parametric test. Wilcox test was used to compare the two groups, and Kruskal test was used for global comparison of multiple groups. Pearson correlation coefficient was used to perform the correlation between immune cells. *p* values < 0.05 was considered statistically significant.

## 3. Results

### 3.1. The ICD-Associated Gene Signature in HGGs

Given that the ICD signature in HGGs remains unclear, we performed bioinformatic analyses using the TCGA database to identify ICD gene metagene [17,18]. The workflow of the study is shown in Figure 1. The expression of these ICD-related genes (33 genes panel) in HGGs samples and the corresponding patient survival data were used in gene clustering. Clustering numbers (k) between two to nine were evaluated (Appendix A). The classification of glioma samples into two clusters showed an ideal clustering effect (Figure 2A). ICD-related genes were upregulated in Cluster 1 (C1) and were downregulated in Cluster 1 (C2) (Figure 2B). We also found that patients stratified into the C1 group had a worse survival in comparison with those stratified into the C2 group (Figure 2C). The differentially expressed genes (DEG) between the C1 and C2 groups were identified by ICD genotyping (Figure 2D,E).

### 3.2. Potential Biological Functions and Signal Pathways Associated with ICD Signature

To elucidate the potential functions and the underlying pathways associated with ICD gene signature, we performed GO enrichment analysis and KEGG enrichment analysis using DEGs between ICD signature-high group (C1) and ICD signature-low group (C2). GO enrichment analyses showed that DEGs were positively correlated with humoral immune response, immunoglobulin complex, antigen binding and other immune functions (Figure 3A,B). KEGG enrichment analyses indicated that DEGs were positively correlated with cytokine receptor interaction and other signaling associated with immunity (Figure 3D). We also performed KEGG enrichment analysis on C1 genes, and the results showed that enriched ICD signature was associated with cytokine signaling activation (Figure 3C). In addition, the C1 group showed upregulation of adaptive immune response, complement activation and humoral immune response (Figure 3E). C2 group, on the other hand, the results showed reduced immunological activity and were enriched by other pathways associated with tumor progression or neurological functions, including DNA conformation change, intrinsic component of synaptic membrane and neuron-to-neuron synapse (Figure 3F).

### 3.3. Upregulated ICD Signature Was Correlated to Reduced Somatic Tumor Mutations and High Infiltration of Immune Cells in HGGs

To further explore the value of ICD genotyping, we analyzed the gene mutation frequencies between ICD-differential C1 and C2 groups. Our data identified reduced tumor gene mutation frequency in the C1 group relative to the C2 group, including TP53 (32% in C1 vs. 34% in C2), EGFR (16% in C1 vs. 34% in C2), TTN (27% in C1 vs. 28% in C2) and MUC16 (14% in C1 vs. 17% in C2) (Figure 4A). We also analyzed the association between ICD gene signature and tumor immunity in HGGs. We found that ICD signature-high C1 showed high ESTMATES score, ImmuneScore and StromalScore, supporting that upregulated ICD signature was associated with increased immune cell infiltration (Figure 4C–E). In contrast, ICD signature-high C1 showed a reduced Tumor Purity Score (Figure 4F). To evaluate the relationship between ICD gene signature and immune cell proportion, the scoring of immune cell components was analyzed in HGG samples using bulk RNA-seq data (Figure 4G). In silico analyses showed that increased ICD signature was correlated with high infiltration of gamma delta T cells and plasma cells but was negatively associated with the proportion of infiltrating monocytes and M0 macrophages (Figure 4H). Meanwhile, the frequency of memory B cells and CD4 memory T cells were upregulated in the C1 group relative to the C2 group (Figure 5A). Furthermore, immune checkpoint-related genes (CD274, SIGLEC15, CTLA4, LGA3, PDCD1LG2, PDCD1, TIGIT, HAVCR2), as well as most of the HLA family genes, were also upregulated in C1 HGGs samples (Figure 5B,C). To determine the prognostic value of the ICD gene signature, we performed univariate cox analysis using the TCGA HGGs cohort and found that increased ICD genes were associated with poor prognosis of HGGs patients (Figure 5D).

### 3.4. Construction of ICD Risk Signature Model and Its Value in HGGs Patient Prognosis Prediction

To evaluate the prognostic value of ICD genes in HGGs, 33 ICD genes were used to construct a prognostic signature model. The prognostic signature model was constructed by LASSO regression using TCGA HGGs dataset as a training cohort (Figure 6A,B) and was validated using the CGGA HGGs dataset. Kaplan–Meier survival analysis showed that patients with high ICD risk scores had poor outcomes (Figure 6C–F). Furthermore, FOXP3, IL6 LY96, MYD88 and PDIA3 used for the ICD risk signature construction were upregulated in ICD high-risk group. The prognostic value of the ICD risk signature was confirmed in both TCGA and CGGA samples using univariate and multivariate cox regression analysis (Figure 7A,B). We also performed an ROC curve to evaluate the prognostic value of the ICD risk signature model. The AUC values of the ICD risk signature model for the TCGA HGGs patients at one year, three years and five years were 0.664, 0.775 and 0.879, respectively (Figure 7C). Similarly, the AUC values of the ICD risk signature model for the CGGA HGGs patients at one year, three years and five years are 0.731, 0.773 and 0.771, respectively (Figure 7D). The AUC value of the ICD risk score was higher than that of other clinical prognostic characteristics, including age, gender, pathological grade, IDH mutation, 1p19q co-deletion and MGMT methylation (Figure 7E,F). Additionally, we detected the survival of patients with the clinical prognostic characteristics respectively in TCGA and CGGA. As shown in Appendix A, the patients of age ≤ 65 and different gender in high risk had poor prognoses in TCGA and CGGA. Furthermore, the patients of WHO 2-3 (recurrent WHO 2, AA WHO), WHO 4 (GBM), IDH status, MGMT methylation status and 1p19q non-co-deletion in high-risk had poor prognosis in CGGA simultaneously.

### 3.5. High ICD Gene Signature Is Associated with Reduced ICB Immunotherapy Response

We then determine whether the ICD gene signature could predict the therapeutic efficacy of ICB treatment. We found that the high ICD score was associated with a reduced proportion of eosinophils and an increased number of regulatory T cells (Tregs) (Figure 8A,B). Interestingly, HGG patients that do not respond to ICB therapy had a high ICD gene signature (Figure 8C). HGGs patients with high ICD gene signatures are associated with high immuno-escape scores (Figure 8D). These data suggest that a high ICD gene signature was associated with reduced ICB immunotherapy response.

## 4. Discussion

WHO grade 3–4 HGGs, are the most prevalent malignant neuroepithelial tumors, comprising astrocytoma, IDH-mutant, grade 3–4; oligodendroma, IDH-mutant and 1p/19q co-deletion, grade 3; GBM, IDH-wild type, grade 4. Astrocytoma, IDH-mutant, grade 3–4 may be developed from a low-grade malignant form [30]. The prognosis of HGGs is poor after maximal surgical resection followed by radiotherapy plus chemotherapy [31]. Immunotherapeutic approaches focusing on enhancing autonomous immunity or passively introducing exogenous anti-tumor immune cells are under intensive investigation and show promising outcomes in preclinical assays and in several clinical trials [32]. Most HGGs belong to “cold tumors” that lack tumor-infiltrating lymphocytes (TILs). How to make these “cold tumors” hot serve as a bottleneck and as a potentiating factor for the development of immunotherapy strategies [33].

ICD is induced by chronic exposure to damage-associated molecular patterns (DAMPs) and may attribute to a dysfunctional anti-tumor immune system [6,9,10]. It might be beneficial to identify ICD-related biomarkers for HGG patients. In this study, we demonstrate a genotyping of ICD which is closely associated with the prognosis in HGGs. The results showed that patients with high expression of ICD genes (C1) had worse survival. Furthermore, we found several genes, such as *IL31RA*, *PAEP* and *LINC00973*, were up-expressed in the C1 group. Whereas the ICD signature represents a prognostic indicator of better patient outcomes in many kinds of cancer [11,12], in HGGs, a high ICD signature is associated with worse patient outcomes.

Possible reasons for the results above could be the restriction to a few ICD genes and the cellular heterogeneity related to cancer types. HGGs, exhibit significant inter-tumoral and intra-tumoral heterogeneity, making the development of effective therapeutic strategies complicated [34]. We found that ICD genotyping correlates with immune response function and signaling, abundant immune cell infiltration, high-levels of immune checkpoint and HLA-related genes, which is inconsistent with previous studies [18,35]. We also developed a prognostic gene signature model containing *FOXP3*, *IL6 LY96*, *MYD88* and *PDIA3* using the TCGA and CGGA datasets.

Forkhead box P3 (FOXP3)^+^ tumor-infiltrating lymphocytes (TILs) were associated with tumor angiogenesis and tumor progression in glioma patients [36]. GBM patients with a higher density of FOXP3^+^ TILs showed a relatively poor prognosis [37]. Interleukin 6 (IL6) signaling contributes to glioma aggressiveness [38]. IL6 expression has been reported to be increased upon hypericin-mediated photodynamic therapy and may enhance the ICD process [39]. The role of IL6 in regulating ICD in HGGs may be therapy-related and rely on cell-extrinsic niche factors. Lymphocyte antigen 96 (LY96 or MD2), which is from the toll-like receptor 4 (TLR4) signaling pathway, promotes the development of immunotherapy and targeted therapy of malignancies in tumor occurrence and progression [40]. However, other studies have pointed out that LY96 inhibition would prevent TLR4-mediated inflammatory responses and metastatic cancer growth [41,42]. The potential role of LY96 of ICD in HGGs is still needed to determine in further studies. Curcumin may exert its anti-tumor effects in glioma cells by inhibiting the TLR-4/MYD88 pathway and inducing tumor cell apoptosis [43]. Myeloid differentiation primary response 88 (MYD88) has predictive prognostic value by influencing tumor-infiltrating immune cell dysregulation, especially the M2-type macrophages in glioma patients [44]. Similar to FOXP3, MYD88 may play a potential role in the occurrence and development of HGGs, which make the prognosis worse for the patients. Reducing protein disulfide isomerase family A member 3 (PDIA3) expression in GBM cells significantly limited the microglia pro-tumor polarization toward the M2 phenotype and the production of pro-inflammatory factors [45]. PDIA3 acts as a robust tumor biomarker in influencing protein synthesis, degradation or secretion and then shaping the tumor microenvironment [46].

There are several limitations of this study. Only ICD gene metagene from the previous studies was used in this study. The RNA-seq data of adjacent brain tissues of glioma in the open-access databases are not sufficient, and we have not performed DEGs between tumor and para-tumor tissues. The functional significance of the presumed ICD gene signature was not investigated in this study. The detailed immune processes associated with the ICD gene signature remain largely unknown. Further translational studies are warranted to verify the clinical significance of ICD in HGGs.

## 5. Conclusions

In the present study, we establish a new classification system and identify ICD prognostic indicator genes: *FOXP3*, *IL6 LY96*, *MYD88* and *PDIA3* that inform poor HGG patient prognosis. This study provides new perspectives and insights and serves as a reminder for HGG patients undergoing ICD-associated immunotherapy.

## Figures and Tables

**Figure 1 cells-11-03655-f001:**
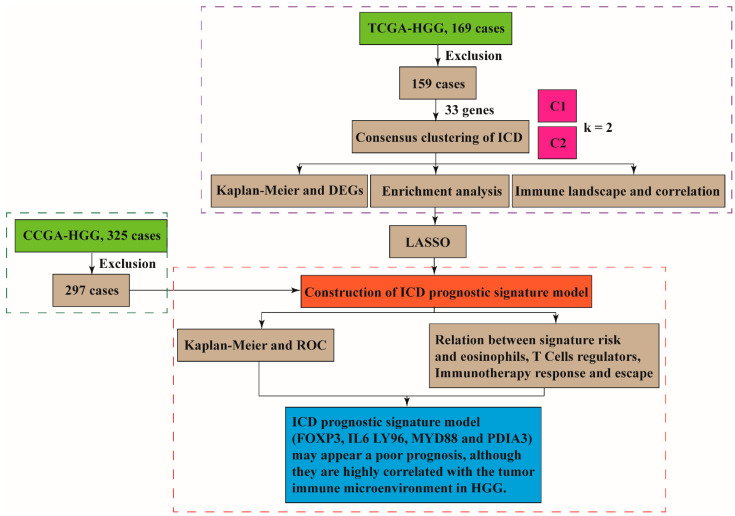
The workflow of the study.

**Figure 2 cells-11-03655-f002:**
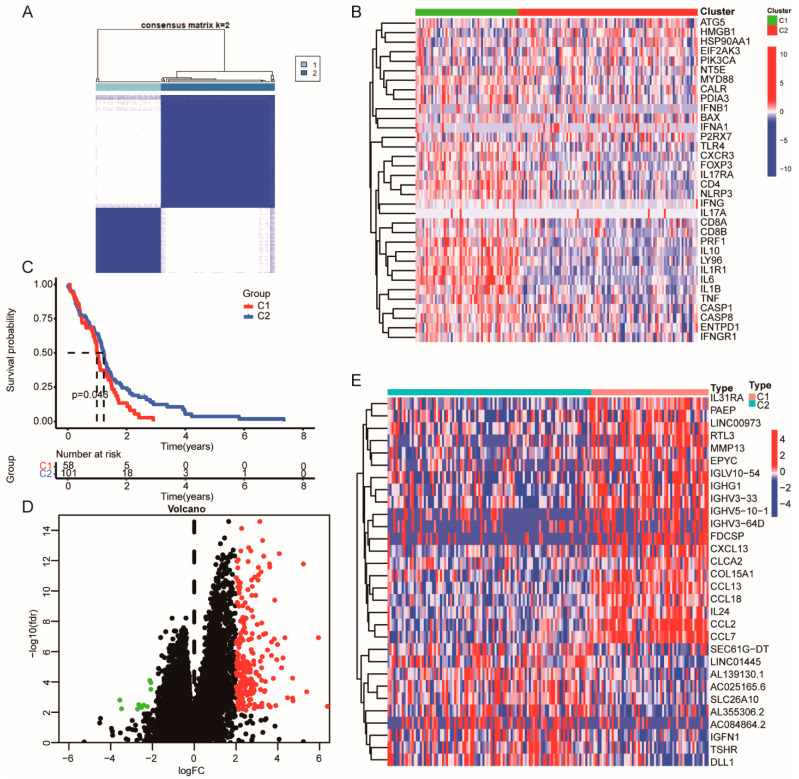
Identification of ICD genotyping by consensus clustering in HGGs. (**A**) Heatmap of the ideal consensus clustering solution (k = 2) for 33 genes in glioma samples. (**B**) Heatmap of differential ICD genes in different ICD genotypes. (**C**) Survival analysis of patients in different ICD genotypes. (**D**,**E**) Volcano plot and heatmap of DEGs in different ICD genotypes. (green dots mean downregulated genes; black dots mean unaltered genes; red dots mean upregulated genes).

**Figure 3 cells-11-03655-f003:**
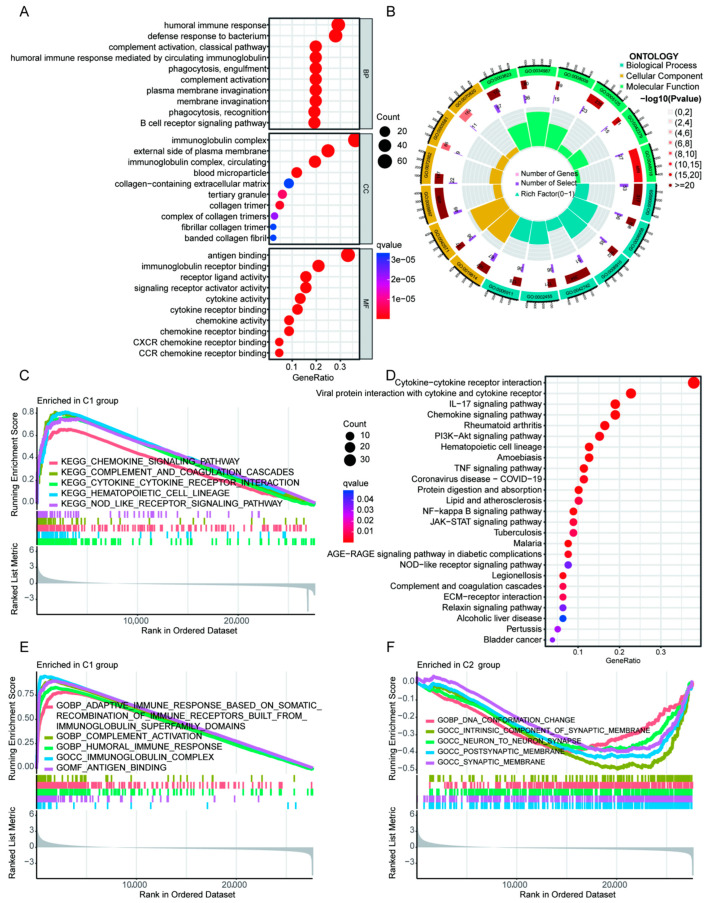
Enrichment analysis of DEGs in ICD genotyping of HGGs. (**A**,**B**) GO enrichment analysis in different ICD genotyping. (**C**) KEGG enrichment analysis in C1 group. (**D**) KEGG enrichment analysis in different ICD genotyping. (**E**,**F**) GO enrichment analysis in C1 and C2 groups, respectively.

**Figure 4 cells-11-03655-f004:**
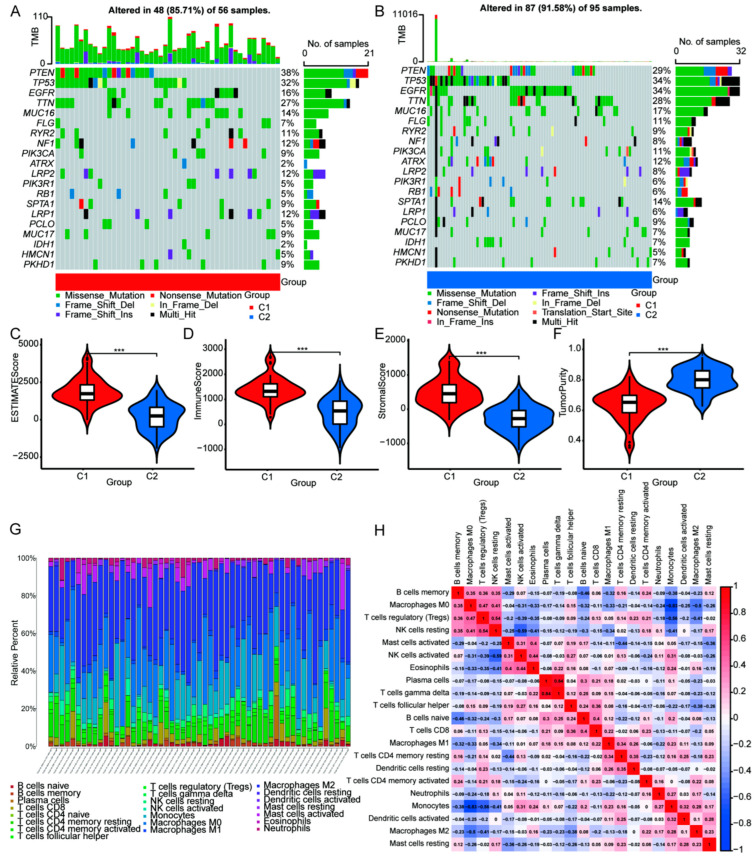
Comparison of somatic mutations, immune landscape and correlation in ICD genotyping of high-grade glioma. The top 20 most frequently mutated genes were oncoprint visualized in C1 (**A**) and C2 group (**B**). ESTMATES score (**C**), ImmuneScore (**D**), StromalScore (**E**), TumorPurity (**F**) in different ICD genotyping. (**G**) The immune cell composition of each glioma sample. (**H**) Correlation analysis of different immune cells in glioma samples (*** *p* < 0.001 vs. indicated group).

**Figure 5 cells-11-03655-f005:**
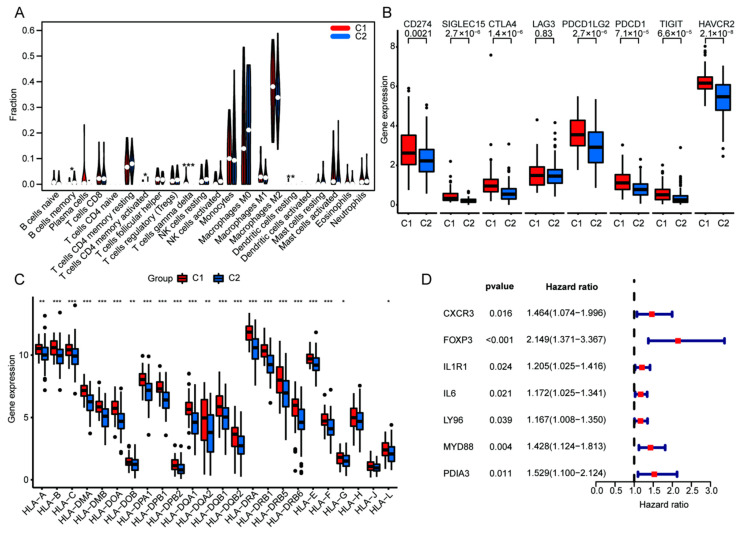
Comparison of TIME in ICD genotyping and revelation of the seven ICD genes with potential prognostic value. (**A**) Violin plot of immune cells in C1 and C2 group. Box plots of immune checkpoints (**B**) and HLA genes (**C**). (**D**) Univariate cox analysis for the prognostic value of the seven ICD genes according to OS. (* *p* < 0.05, ** *p* < 0.01 and *** *p* < 0.001 vs. indicated group, dots mean discrete value).

**Figure 6 cells-11-03655-f006:**
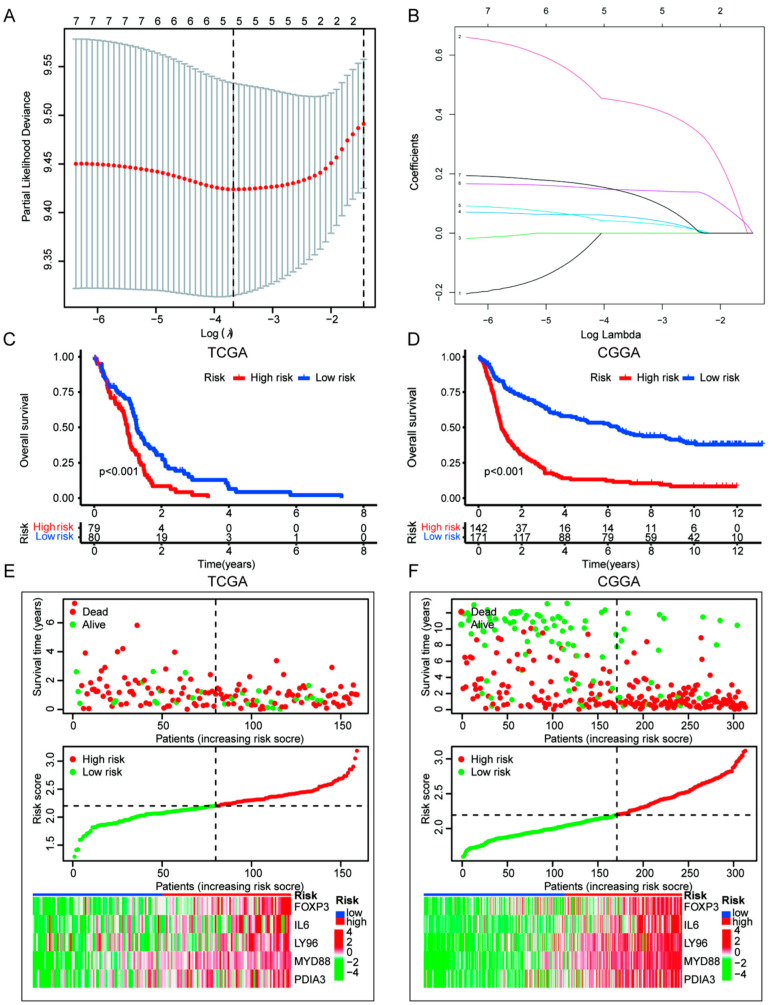
Construction and validation of the ICD risk signature in TCGA and CGGA database. (**A**,**B**) LASSO regression of the most associated genes with OS in TCGA. (**C**,**D**) Kaplan–Meier analyses in TCGA and CGGA databases. (**E**,**F**) Survival status, risk scores distribution, and heatmaps of five prognostic gene signatures of each patient in TCGA and CGGA.

**Figure 7 cells-11-03655-f007:**
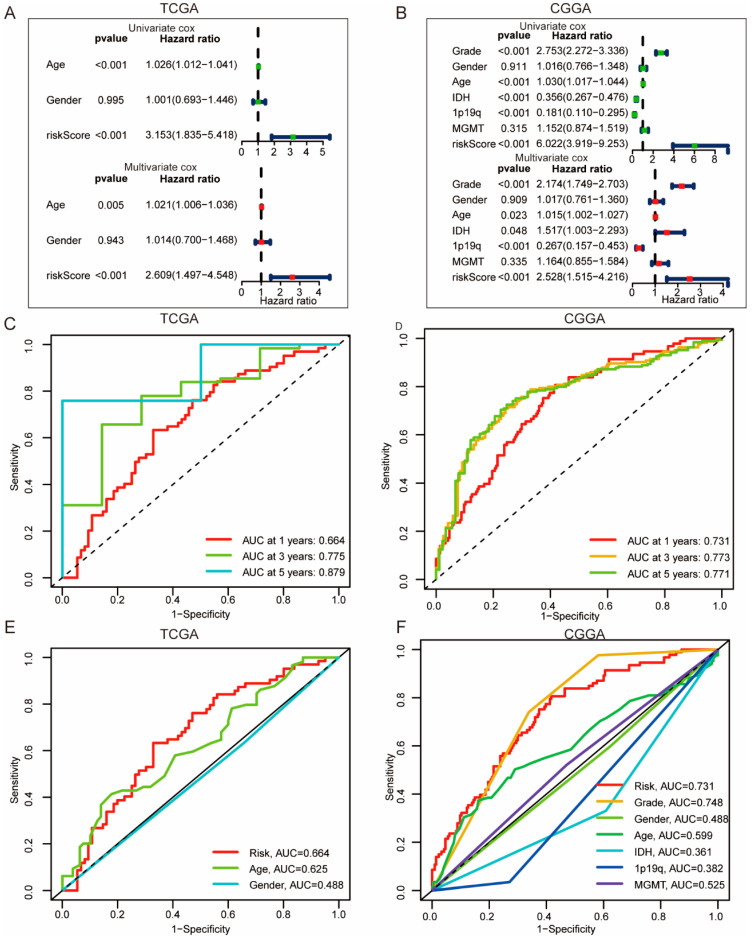
The prognostic value of ICD prognostic signature model in TCGA and CGGA database. The independent prognostic value of ICD risk signature by univariate and multivariate Cox analysis in TCGA (**A**) and CGGA (**B**) database. One-, three- and five-year ROC curves of prognostic value in TCGA (**C**) and CGGA (**D**) database. ROC curves of ICD prognostic risk score, age and grade in TCGA (**E**), and ICD prognostic risk score, age, grade pathological grade, IDH mutation, 1p19q co-deletion and MGMT methylation in CGGA (**F**) database.

**Figure 8 cells-11-03655-f008:**
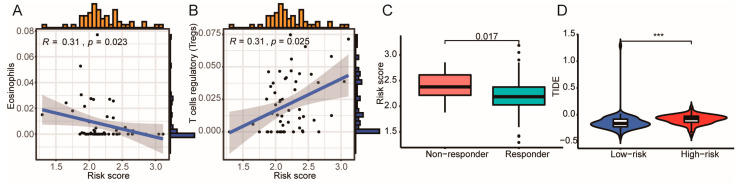
The prognostic value of ICD prognostic signature model and immunotherapy sensitivity. Correlation between ICD prognostic risk score with the content of eosinophils (**A**) and T Cells regulators (**B**). (**C**) Risk scores for the response to ICB immunotherapy in different patients. (**D**) Immunotherapy escape in different patients of ICD prognostic risk scores. (*** *p* < 0.001 vs. indicated group).

**Table 1 cells-11-03655-t001:** Patient clinical information in TCGA and CGGA HGG cohort.

Dataset	TCGA (n = 159)	CGGA (n = 297)
Age (Years)	59.41 ± 13.66	43.31 ± 12.00
Gender (M/F)	100/59	181/116
Grade (2/3/4)	/	91/73/133
IDH mutation (Y/N)	/	140/157
1p/19q co-deletion (Y/N)	/	235/62
MGMT promoter methylation (Y/N)	/	152/145

## Data Availability

Publicly available datasets were as followed: TCGA, https://cancergenome.nih.gov/and CGGA, http://www.cgga.org.cn/ (accessed on 1 September 2022).

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
