# Peer review of "Upregulated Immunogenic Cell-Death-Associated Gene Signature Predicts Reduced Responsiveness to Immune-Checkpoint-Blockade Therapy and Poor Prognosis in High-Grade Gliomas"

_cells, 2022, doi:10.3390/cells11223655_

Round 1

Reviewer 1 Report

Drs Tang et al. wrote a fairly comprehensive in silico analysis (in publicly available databases) of ICD related gene signatures in high grade gliomas. The topic is timely and relevant, gaining more and legitimate attention of the cancer immunotherapy field. Glioblastoma is known to behave notoriously different on several types of immunotherapy as compared to other solid cancers. My main concerns can be summarized in 3 points:

1. High grade gliomas are clinically a very heterogenous group with a spectrum going from anaplastic oligodendrogliomas towards glioblastoma. The outcome under contemporary standard of care therapy is hugely different: this is particularly a problem if you know that especially the IDH mutated tumors ( regardless the pathological grade ) are extremely deprived from immune cell infiltration while they are having a far better prognosis than GBM. The authors included significant numbers of IDH mutated ( and even 1p19q codeleted ) tumors. By including these entities they introduce an enormous bias, skewing the correlations of good outcome towards immune deserted tumors. For the analysis performed in this work, this is problematic. Therefore I would strongly advise the authors to perform exactly the same analyses on the true group of GBM ( defined according to the WHO 2021 criteria ), leaving out all grade III lesions, all IDH mutated and certainly all 1p19q deleted lesions. I really wonder whether the results will still prove to be that clear then.

2. ICD propensity should be a functional definition, rather than merely a gene signature. Of course the Bulk RNA seq analysis that is on the basis of the presumed ICD reactions can be a surrogate to be measured, but this shortcoming should definitively be acknowledged in the text.

3.Throughout the manuscript it is unclear which type of immunotherapy and which type of 'response to immunotherapy' is referred to as an outcome measure: are we talking about Check Point inhibitors ( which don't work in GBM ), adoptive T cell strategy or DC and other types of vaccination or oncolytic virus therapy ? Is a response defined as an objective radiological change in tumor load or rather as a dichotomized survival outcome ? This really is crucial to understand the results.

Minor remarks: the authors should check the references, when cited in the text: e.g. on p.2 "Abhishek D et al" should read " Garg et al."

Author Response

Response to Reviewer 1

Reviewer 1:

Drs Tang et al. wrote a fairly comprehensive in silico analysis (in publicly available databases) of ICD related gene signatures in high grade gliomas. The topic is timely and relevant, gaining more and legitimate attention of the cancer immunotherapy field. Glioblastoma is known to behave notoriously different on several types of immunotherapy as compared to other solid cancers. My main concerns can be summarized in 3 points:

Thank you for giving us the opportunity to submit a revised draft of the manuscript “Classification of Immunogenic Cell Death Associated Genes Emerges Poor Prognosis and Non-response to Immunotherapy in High-grade Glioma” for publication in the Journal of Cells. We appreciate the time and effort that you dedicated to providing feedback on our manuscript and are grateful for the insightful comments on and valuable improvements to our article. We have incorporated according to the suggestions. Those changes are highlighted in the manuscript. Please see below, in red, for a point-by-point response to the reviewers’ comments and concerns. All page numbers refer to the revised manuscript file with tracked changes.

1. High grade gliomas are clinically a very heterogenous group with a spectrum going from anaplastic oligodendrogliomas towards glioblastoma. The outcome under contemporary standard of care therapy is hugely different: this is particularly a problem if you know that especially the IDH mutated tumors ( regardless the pathological grade ) are extremely deprived from immune cell infiltration while they are having a far better prognosis than GBM. The authors included significant numbers of IDH mutated ( and even 1p19q codeleted ) tumors. By including these entities they introduce an enormous bias, skewing the correlations of good outcome towards immune deserted tumors. For the analysis performed in this work, this is problematic. Therefore I would strongly advise the authors to perform exactly the same analyses on the true group of GBM ( defined according to the WHO 2021 criteria ), leaving out all grade III lesions, all IDH mutated and certainly all 1p19q deleted lesions. I really wonder whether the results will still prove to be that clear then.

Response: We feel great thanks for your professional review work on our article. According to your nice suggestions, we have provide the survival of patients with the clinical prognostic characteristics respectively in TCGA and CGGA , especially for WHO 2-3 (recurrent WHO 2, AA WHO 3), WHO 4 (GBM), IDH, MGMT methylation and 1p19q codeletion status. The details were shown in Supplement Figure 2 (Line 227-234).

2. ICD propensity should be a functional definition, rather than merely a gene signature. Of course the Bulk RNA seq analysis that is on the basis of the presumed ICD reactions can be a surrogate to be measured, but this shortcoming should definitively be acknowledged in the text.

Response: We sincerely appreciate the valuable comments, we have acknowledged the shortcoming in Discussion (Line 342-344).

3.Throughout the manuscript it is unclear which type of immunotherapy and which type of 'response to immunotherapy' is referred to as an outcome measure: are we talking about Check Point inhibitors ( which don't work in GBM ), adoptive T cell strategy or DC and other types of vaccination or oncolytic virus therapy ? Is a response defined as an objective radiological change in tumor load or rather as a dichotomized survival outcome ? This really is crucial to understand the results.

Response: Thank you for pointing this out, according to your valuable comments, we have provided the details of TIDE, the type of 'response to immunotherapy' and definition of the immunotherapy response in Prediction of response to immunotherapy (Line 130-134)

Minor remarks: the authors should check the references, when cited in the text: e.g. on p.2 "Abhishek D et al" should read " Garg et al."

Response: We were really sorry for our careless mistakes. Thank you for your reminder. We have revised the name “Garg AD (Garg Abhishek D, Line 59)”.

Based on these comments and suggestions, we have made careful modifications to the original manuscript, and carefully proof-read the manuscript to minimize typographical and grammatical errors. We believe that the manuscript has been greatly improved and hope it has reached your journal’s standard.

Reviewer 2 Report

The authors suggest that ICD signature model has a poor prognosis in HGG.

The following points have to be addressed:

- define more clearly HGG. were anaplastic ologodendroglioma included.

- Did GBM differ from anaplastic astrocytomas?

- Explain how the ICD prognostic signature model should be implemented in the daily routine.

- How is the model correlated with tumor microenvironment.

- what is a cold tumor (line 263)

- There are many language errors

Author Response

Response to Reviewer 2

Reviewer 2:

The authors suggest that ICD signature model has a poor prognosis in HGG.

The following points have to be addressed:

Thank you for giving us the opportunity to submit a revised draft of the manuscript “Classification of Immunogenic Cell Death Associated Genes Emerges Poor Prognosis and Non-response to Immunotherapy in High-grade Glioma” for publication in the Journal of Cells. We appreciate the time and effort that you dedicated to providing feedback on our manuscript and are grateful for the insightful comments on and valuable improvements to our article. We have incorporated according to the the suggestions. Those changes are highlighted in the manuscript. Please see below, in red, for a point-by-point response to the reviewers’ comments and concerns. All page numbers refer to the revised manuscript file with tracked changes.

- define more clearly HGG. were anaplastic ologodendroglioma included.

Response: Thank you for your positive comments on our manuscript. We have revised the definition of HGG in Discussion (Line 45, Line 261-263). According to the WHO 2021 criteria, anaplastic ologodendroglioma was instead by ologodendroglioma WHO Ⅲ. But in the database, anaplastic ologodendroglioma was still included.

- Did GBM differ from anaplastic astrocytomas?

Response: Thank you for your nice comments on our article, anaplastic astrocytomas without CDKN2A/B homozygous deletion is belong to WHO 3, but GBM is always considered WHO 4, according to the WHO 2021 criteria.

- Explain how the ICD prognostic signature model should be implemented in the daily routine.

Response: We sincerely appreciate the valuable comments. It would have been interesting to explore this aspect. ICD prognostic signature model is established based on gene expression profile. Therefore, tissue or blood samples of patients may be collected for gene expression determination in subsequent clinical work, such as ELISA, PCR, IHC, WB, etc. Perhaps a gene panel can be suggested to assist in predicting the prognosis of high-grade gliomas.

- How is the model correlated with tumor microenvironment.

Response: We think this is an excellent suggestion. Please allow me to explain this:

  1. ICD associated genes are mainly involved in tumor immune mechanism, and immune cells are mainly located in tumor microenvironment.
  2. We found that ICD genotyping was correlated with the content of some immune cells, immune checkpoint genes and HLA family genes.
  3. In this study, we analyzed the strong correlation between risk score and immune cells, such as Eosinophils and Tregs, as well as the relationship between risk score and immunotherapy response and immune escape.

- what is a cold tumor (line 263)

Response: Thank you for your nice comments on our article. According to your suggestions, we have explain the “cold tumor” in the Discussion (Line 269-271)

- There are many language errors

Response: We tried our best to improve the manuscript and made some changes to the manuscript. These changes will not influence the content and framework of the paper. And here we did not list the changes but marked in red in the revised paper. We invited a friend who is a native English speaker from USA to help polish our article, including the title, abstract and full text. And we hope the revised manuscript could be acceptable for you.

Based on these comments and suggestions, we have made careful modifications to the original manuscript, and carefully proof-read the manuscript to minimize typographical and grammatical errors. We believe that the manuscript has been greatly improved and hope it has reached your journal’s standard.

Round 2

Reviewer 1 Report

The authors partially complied with the concerns made in the first review process. Supplementary figure S2 is a crucial finding and should get more visibility in the main text.

The main concern still remaining is that the authors don't provide the definition of what, for them, is defined as "response" and "no response" to immunotherapy. Are these terms to be read as "longer survival" and "shorter survival" OR rather as "objective response documented in patient" and "no objective response documented in patient". It really is cut too short to call the groups with a better survival "responders on immunotherapy" since ICB will only have been a minor part of the therapies administered to these patients in different clinical trial designs. The survival should be an ultimate result of all therapies applied and my personal hypothesis is that ICB don't have anything to do with the documented survival ratios in the different groups analysed. So, a clear definition of what is considered "response" and "no response" really has to be provided and is vital for correct interpretations of the in silico analysis.

Author Response

Reviewer 1

The authors partially complied with the concerns made in the first review process. Supplementary figure S2 is a crucial finding and should get more visibility in the main text.

The main concern still remaining is that the authors don't provide the definition of what, for them, is defined as "response" and "no response" to immunotherapy. Are these terms to be read as "longer survival" and "shorter survival" OR rather as "objective response documented in patient" and "no objective response documented in patient". It really is cut too short to call the groups with a better survival "responders on immunotherapy" since ICB will only have been a minor part of the therapies administered to these patients in different clinical trial designs. The survival should be an ultimate result of all therapies applied and my personal hypothesis is that ICB don't have anything to do with the documented survival ratios in the different groups analysed. So, a clear definition of what is considered "response" and "no response" really has to be provided and is vital for correct interpretations of the in silico analysis.

Response: Thank you again for your nice comments on our article. According to your suggestions, we have checked the relevant references and TIDE website. We apologized for our general judgment of the ICB response as an immune response. And we have revised all the “response to immunotherapy” to “response to ICB immunotherapy”, and cited these references.

TIDE web platform to infer gene functions in modulating tumor immunity and evaluate biomarkers to predict ICB response, and its module displays the comparison between the custom biomarker and other published biomarkers based on their predictive power of response outcome and overall survival [1]. In the cytotoxic T lymphocyte (CTL)-high tumors, TIDE correlates the tumor expression data with the T cell dysfunction signature and predicts tumors with high correlation to T cell dysfunction as non-responders. In CTL-low tumors, it has been reported that ICB can enhance the cytotoxic T cell infiltration [2].

On the TIDE web, Predicted responder predictions by the threshold of the TIDE score set by a user (default is 0). The TIDE score means “response” to ICB when represent positive value, otherwise, “no response”. We have revised the definition of what is considered "response" and "no response" in “2.6. Prediction of response to ICB immunotherapy”.

References

[1]Fu J, Li K, Zhang W, Wan C, Zhang J, Jiang P, Liu XS. Large-scale public data reuse to model immunotherapy response and resistance. Genome Med. 2020, 12(1):21.

[2]Jiang P, Gu S, Pan D, Fu J, Sahu A, Hu X, Li Z, Traugh N, Bu X, Li B, Liu J, Freeman GJ, Brown MA, Wucherpfennig KW, Liu XS. Signatures of T cell dysfunction and exclusion predict cancer immunotherapy response. Nat Med. 2018,24(10):1550-1558.

Reviewer 2 Report

Not much has been improved since the last submission.

Author Response

Reviewer 2

Not much has been improved since the last submission.

Response: Thanks for your suggestion. We feel sorry for our poor writings, however, we tried our best to polish the language in the revised manuscript. We have almost reedited the text of the whole article.

And we hope the revised manuscript could be acceptable for you.